# Induction of Skin Cancer by Long-Term Blue Light Irradiation

**DOI:** 10.3390/biomedicines11082321

**Published:** 2023-08-21

**Authors:** Keiichi Hiramoto, Sayaka Kubo, Keiko Tsuji, Daijiro Sugiyama, Hideo Hamano

**Affiliations:** 1Department of Pharmaceutical Sciences, Suzuka University of Medical Science, Suzuka 513-8670, Japan; 2Research Department, Daiichi Sankyo Healthcare Co., Ltd., Chuo-ku, Tokyo 140-8170, Japan; kubo.sayaka.hr@daiichisankyo-hc.co.jp (S.K.); tsuji.keiko.nj@daiichisankyo-hc.co.jp (K.T.); sugiyama.daijiro.gz@daiichisankyo-hc.co.jp (D.S.); hamano.hideo.gg@daiichisankyo-hc.co.jp (H.H.)

**Keywords:** blue light, neutrophil, macrophage, skin cancer

## Abstract

Presently, people are not only exposed to sunlight but also to a large amount of blue light from personal computers and smartphones. This blue light has various effects on the living body. However, its effect on the induction of skin cancer is unknown. In this study, we investigated the induction of skin cancer by long-term blue light irradiation. Hairless mice were irradiated with blue light (LED; peak emission 479 nm) every day for one year, and a control was irradiated with white light (LED), green light (LED; peak emission 538 nm), and red light (LED; peak emission 629 nm) for one year, respectively. Skin cancer was induced only in the mice exposed to blue light. Long-term blue light irradiation also increased the migration of neutrophils and macrophages involved in carcinogenesis in the skin. In neutrophils, an increased expression of citH3 and PAD4 was observed, suggesting the possibility of NETosis. Conversely, in macrophages, inflammatory macrophages (type 1 macrophages) increased and anti-inflammatory macrophages (type 2 macrophages) decreased due to continuous blue light irradiation. These findings suggest that long-term continuous irradiation with blue light induces neutrophil NETosis and an increase in type 1 macrophages, resulting in skin cancer.

## 1. Introduction

Blue light with a wavelength of 380 to 500 nm has the shortest wavelength and highest amount of energy among the visible light seen by the human eye [1]. The LED displays and LED lighting of computers and smartphones contain a lot of blue light [2,3]. Blue light is also involved in the biological clock [4].

The retina contains photoreceptor cells that control circadian rhythms [5]. These cells only respond to light with a wavelength of 460 nm, which is the same wavelength as that of blue light, and exposure to blue light in sunlight during the day helps regulate the internal clock [4,6]. When the body clock is disturbed, sleep disorders occur, affecting the health [7]. Therefore, it can be said that looking at blue light during the day is essential for human life; however, prolonged use of a computer or smartphone regardless of the time of the day, as in modern society, may cause health problems [8]. Further, blue light puts additional strain on the eyes when looking at computers, smartphones, and TVs for long periods of time. As a result, symptoms such as dry eye, blurred vision, and loss of focus appear [9]. When it gets worse, eyestrain, stiff shoulders, and headaches also occur [10].

In general, ultraviolet rays have the highest energy in sunlight and are known to have adverse effects on the eyes and skin [11]. However, ultraviolet light is absorbed by the lens and vitreous body of the eyeball and hardly reaches the retina [12]. Conversely, blue light has the shortest wavelength and strongest energy of visible light. It can reach the retina without being absorbed by the lens or vitreous body together with other light in the long wavelength band [13]. Therefore, exposure to blue light for long periods of time can damage the retina and cause age-related macular degeneration [14].

Further, blue light not only affects the eyes but also the entire body. As mentioned above, prolonged exposure to blue light at night can disturb the biological clock’s distinction between day and night, causing circadian rhythm disturbances. This causes abnormalities in the nervous system, causing internal organs and blood vessels to move actively at night as well as during the day, leading to poor physical condition, the exacerbation of lifestyle-related diseases, and accelerated aging [15]. Also, in the skin, photoaging consists of a decrease in skin elasticity, causing small pimples, lentigines, and mottled pigmentation. Blue light has been shown to induce photoaging in vitro and in vivo, specifically causing oxidative stress, damaging the skin barrier and promoting persistent skin pigmentation [16,17,18].

Recently, many studies have reported the effects of blue light on the eyes and whole body [19,20]. However, no study has investigated the effect of blue light on skin cancer. The purpose of this study was to elucidate the induction of skin cancer and its mechanism by long-term blue light irradiation of the skin of hairless mice.

## 2. Materials and Methods

### 2.1. Animal Experiments

Specific pathogen-free 8-week-old male hairless mice (Hos:HR-1) were obtained from SLC (Hamamatsu, Shizuoka, Japan). The mice were individually bred in cages in an air-conditioned room at 23 ± 1 °C under specific pathogen-free conditions with a 12 h light/12 h dark cycle under stress-free conditions. The mice were quarantined for the first seven days and then randomly assigned to groups of five according to body weight as follows: control, white-light-irradiated, blue-light-irradiated, green-light-irradiated, and red-light-irradiated groups. A fluorescent lamp was used as the light source, with LED blue light (wavelength: 380–500 nm, peak emission: 479 nm, 40 kJ/m^2^, ISLM-150X150-BB, CCS Inc., Kamikyo-ku, Kyoto, Japan), LED green light (wavelength: 500–560 nm, peak emission: 538 nm, 40 kJ/m^2^, ISLM-150X150-GG, CCS Inc.), LED red light (wavelength: 600–700 nm, peak emission: 629 nm, 40 kJ/m^2^, ISLM-150X150-RR, CCS Inc., Kamikyo-ku, Kyoto, Japan), or LED white light (12 kJ/m^2^, ISLM-150X150-HWW, CCS Inc., Kamikyo-ku, Kyoto, Japan). The energy content of each LED light was measured using a light analyzer LA-105 (Nippon Medical & Chemical Instruments Co., Ltd., Osaka, Japan). In addition, the control group was irradiated with the fluorescent lamp normally used for breeding. The entire bodies of the mice were exposed to each LED light daily (10 min/day) for a period of one year. After collecting tumor data (the mean number of tumors per mouse) on the final day of examination, we extracted blood and skin samples under anesthesia. This study was approved by the Suzuka University of Medical Science Animal Experiment Ethics Committee on 25 September 2014, and it was performed in strict accordance with the recommendations of the Guide for the Care and Use of Laboratory Animals of the Suzuka University of Medical Science (Approval number: 34). All surgeries were performed on mice under pentobarbital anesthesia, and efforts were made to minimize animal suffering.

### 2.2. Preparation and Staining of Dorsal Skin

On the final day of this experiment, we extracted skin samples under anesthesia. The dorsal skin specimens were fixed in 4% phosphate-buffered paraformaldehyde, embedded in frozen Tissue Tek OCT compound (Sakura Finetek, Tokyo, Japan), and cut into 5 μm thick sections. The sections were then stained with hematoxylin-eosin, in accordance with the established procedures, to enable histological analysis of the skin. Collagen expression was evaluated using the Masson trichrome technique (trichrome stain kit (modified Masson’s); ScyTec Laboratories, Inc., Logan, UT, USA) [21]. Further, the skin specimens were stained using antibodies for immunohistological analysis according to a previously published method [22]. The skin specimens were reactive with either mouse monoclonal anti-lymphocyte antigen 6 complex locus G6D (Ly6G: marker of neutrophil) (1:100; BD Biosciences, Franklin Lakes, NJ, USA), rabbit polyclonal anti-chemokine CX3C receptor 1 (CX3CR1) (1:100; Bioss Antibodies Inc., Woburn, MA, USA), rabbit polyclonal anti-citrullinated histone H3 (citH3) (1:100; Abcam, Cambridge, MA, USA), or rabbit polyclonal anti-protein arginine deiminase 4 (PAD4) (1:100; Abcam) primary antibodies. The samples were then washed and incubated with fluorescein iso-thiocyanate-conjugated anti-mouse and anti-rabbit (1:30; Dako Cytomation, Glostrup, Denmark) secondary antibodies, respectively. The expressions of Ly6G, CX3CR1, citH3, and PAD4 were immunohistochemically evaluated via fluorescence microscopy. The number of Ly6G per mm^2^, CX3CR1 per mm^2^, citH3 per mm^2^, and PAD4 per mm^2^ was then determined by counting 10–15 fields at random at 9200 magnifications. Additionally, collagen type I was calculated from five random visual fields with a constant area using Image J software ver. 1.53 (National Institutes of Health, Bethesda, MD, USA). Briefly, original files were converted to monochrome 8-bit files. Next, the threshold of luminous intensity was voluntarily established. Areas above the threshold were measured in each sample. These areas were defined as “intensity” in this study.

### 2.3. Western Blotting of Dorsal Skin Proteins

The dorsal skin samples were homogenized in lysis buffer (Kurabo, Osaka, Japan). Following this, the homogenates were centrifuged at 8000× *g* for 10 min, and the resultant supernatants were collected. Western blotting was performed as previously described [23]. Briefly, the membranes were incubated with primary antibodies against F4/80 (marker of macrophage; 1:1000; ThermoFisher Scientific, Waltham, MA, USA), chemokine receptor 7 (CCR7: marker of M1 type macrophage; 1:1000; Abcam, Cambridge, MA, USA), CD163 (marker of M2 type macrophage; 1:1000; ThermoFisher Scientific), or β-actin as a loading control (1:5000; Sigma-Aldrich, St. Louis, MO, USA) at room temperature for 1 h. The membranes were then washed and incubated with horseradish peroxidase-conjugated secondary antibody (Novex, Frederick, MD, USA). Immune complexes were detected using ImmunoStar Zeta reagent (Wako, Osaka, Japan), and images were acquired using Multi Gauge software ver. 3.0 (Fujifilm, Greenwood, SC, USA).

### 2.4. Measurement of IL-10, TGF-β, IL-6, and IL-23 Levels in Plasma and Ki67, Cyclin D1, Neutrophil Elastase, and Reactive Oxygen Species Levels (ROS) in Dorsal Skin

The blood and dorsal skin samples were collected on the last experimental day. The plasma was separated from the blood samples via centrifugation at 3,000× *g* and 4 °C for 10 min, and the supernatant was used for further analysis. The levels of IL-10, TGF-β, IL-6, and IL-23 were determined using commercial enzyme-linked immunosorbent assay kits (IL-10, IL-6, and IL-23; MyBiosource, San Diego, CA, USA; TGF-β; Promega, Madison, MI, USA) as per the manufacturer’s instructions. We first rinsed 10 mg of dorsal skin in phosphate-buffered saline to remove excess blood before homogenization. The samples were then homogenized at 1500 × *g* and 4 °C for 15 min, and the supernatant was collected for assaying. The levels of Ki67, cyclin D1, and neutrophil elastase in the dorsal skin were determined using commercial enzyme-linked immunosorbent assay (ELISA) kits (Ki67; Wuhan Fine Biotech, Hubei, China; cyclin D1; Novus, Centennial, CA, USA; neutrophil elastase; R&D Systems, Inc., Minneapolis, MN, USA) according to the manufacturers’ instructions. The ROS levels in the dorsal skin were determined with an OxiSelect^TM^ STA-347 in vivo ROS/RNS assay kit (Cell Biolabs, Inc., San Diego, CA, USA) in accordance with the manufacturer’s instructions. Optical density was measured using a microplate reader (Mdecular Devices, Sunnyvale, CA, USA).

### 2.5. Statistical Analysis

All data are presented as the mean ± standard deviation. Microsoft Excel 2010 (Microsoft Corp., Redmond, WA, USA) was used to analyze the statistical significance of the data along with one-way analysis of variance followed by Tukey’s post hoc test using SPSS version 20 (SPSS Inc., Chicago, IL, USA). *p* values * < 0.05 and ** < 0.01 were considered significant.

## 3. Results

### 3.1. Effects of Blue Light on Skin Cancer

After daily exposure to blue light for one year, skin cancer was observed microscopically (Figure 1A–C). Blue light irradiation increased the expression of ki-67 and cyclin D1, which are indicators of cancer cell proliferation (Figure 1D,E). Conversely, white, green, and red light did not induce skin cancer.

### 3.2. Effects of Blue Light on Total Skin Collagen

The dermal expression of collagen decreased in blue-light-irradiated mice compared to that in the control mice (Figure 2). Under white light, green light, and red light, collagen expression in the skin did not differ from that of the control group.

### 3.3. Effects of Blue Light on Expression of Neutrophil-Related Substances

First, we examined the neutrophil-related substances involved in cancer induction. Neutrophil expression increased following blue light irradiation (Figure 3B). The expression of CX3CR1, a chemotactic factor for neutrophils, was also increased by blue light irradiation (Figure 3C). Additionally, we observed the co-localization of neutrophils and CX3CR1. The blue-light-treated group showed increased co-localization of neutrophils and CX3CR1 (Figure 1A). The expression of citH3 and PAD4, which are indicators of NETosis, increased in the blue-light-irradiated group (Figure 3D,E). Further, the expression of neutrophil elastase and ROS, which are involved in NETosis, also increased in the blue light irradiation group (Figure 3F,G). The white-light-, green-light-, and red-light-irradiated groups did not differ from the control group.

### 3.4. Effects of Blue Light on Expression of Macrophage-Related Substances

Next, we examined macrophage-related substances. The expression of macrophages increased in the blue light irradiation group (Figure 4A). The expression of type 1 macrophages, which are inflammatory macrophages, increased following blue and green light irradiation (Figure 4B). Conversely, the expression of anti-inflammatory type 2 macrophages decreased in the blue and green light irradiation groups (Figure 4C). A difference in molecular weight was observed by inserting a molecular weight marker in the Western blot figures of CCR7 and CD163 (Figure 4D). Additionally, the expression of IL-6 and IL-23 secreted by M1 macrophages increased after blue light irradiation (Figure 4G,H), but that of IL-10 and TGF-β secreted by M2 macrophages did not change (Figure 4E,F).

## 4. Discussion

In this study, skin cancer was induced by long-term blue light irradiation of the skin daily. Conversely, white, green, and red light did not cause skin cancer. Blue light irradiation increased the expression of neutrophils and CXCR1, a neutrophil chemotactic factor. Additionally, the expression of CitH3 and PAD4, indicators of neutrophil-induced NETs, increased, and the levels of neutrophil elastase and ROS secreted by neutrophils also increased. Furthermore, irradiation with blue light increased the number of M1-type macrophages and levels of IL-6 and IL-23 in the blood.

In general, when the skin is exposed to UV rays and other stimuli, neutrophils in blood vessels gather at the stimulated site and excessively secrete neutrophil elastase, which decomposes components, such as collagen and elastin, that make up the dermis [24]. In the present study, the expression of CXCR1, a neutrophil chemotactic factor, and neutrophils was also increased by blue light irradiation. Neutrophils migrate to the local area in response to inflammatory stimuli and protect the body from external enemies via phagocytosis and degranulation [25]. Further, activated neutrophils cause extracellular neutrophil traps (NETs), indicating an important role in immune function [26]. The formation of NETs is characterized by the collapse of the neutrophil nuclear membrane, followed by chromatin swelling and cell membrane rupture; this series of cell death processes is called NETosis [27]. The citrullination of histones (citH3), which are important proteins for NETosis, forms NETs by converting arginine to citrulline by the action of peptide deiminase (PAD4) [27]. In this study, the expression of citH3 and PAD4 increased, suggesting an increase in the formation of NETs. NETs are a type of host defense response, but they also act as damage-associated molecular patterns that activate the complement system and inflammasomes and amplify inflammation [28]. Recently, NETs have been reported to be involved in cancer progression and metastasis. NETs are involved in cancer cell invasion and metastasis by inducing epithelial–mesenchymal cell differentiation and transformation through the activation of the EGFR/ERK pathway [29,30]. They also stimulate the proliferation of cancer cells [31] and activate dormant cancer cells [30]. Thus, it was suggested that blue light irradiation activates NETs and causes carcinogenesis and proliferation.

Conversely, in this study, we observed an increase in macrophages following blue light irradiation, indicating that these macrophages are of the M1 type. M1-macrophages secrete inflammatory cytokines. Among them, IL-6 induces carcinogenesis through STAT3 signaling [32]. Furthermore, IL-23 has been shown to contribute to carcinogenesis [33]. However, M1 macrophages generally act as cancer suppressors. In addition, when carcinogenesis occurs due to macrophage-derived humoral factors, the microenvironment surrounding cancer shifts from Th1 cells to a state in which Th2 cells are dominant and macrophages shift to the M2 type accordingly [34,35]. M2 macrophages secrete TGF-β and IL-10 to suppress anti-tumor immunity and induce angiogenesis [36,37], thereby shifting to an environment favorable for cancer cell proliferation.

In addition, blue light inhibits fibroblast proliferation and induces the suppression of type I collagen expression and upregulation of MMP-1 [16,17,18]. And JUN, TGF, and EGFR signaling pathways have been reported to influence blue-light-induced skin aging [38]. In this study, immune cells such as neutrophils and macrophages were also involved in this transduction pathway, and it was thought that a more complex network was formed.

Furthermore, in this study, the amount of ROS produced was increased by blue light irradiation. ROS play an important role in tumor growth. Neutrophils produce ROS through the activation of nicotinamide adenine dinucleotide phosphate (NADPH) oxidase. NADPH oxidase consists of two subunits, p22^Phox^ and gp91^phox^, and contains all the electron transport chains necessary for the production of superoxide anion (O_2_^−^) from NADPH, and converts O_2_^−^ to hydrogen peroxide (H_2_O_2_) and singlet molecular oxygen (^1^O_2_) [39,40]. This suggests that blue light may activate NADPH oxidase. Macrophages are the first immune cells that exert direct anti-bacterial effects by producing ROS [41]. In macrophages, brain-specific angiogenesis inhibitor 1 (BAI1), an adhesion family member of G protein-coupled receptors (GPCR)s, promotes O_2_^−^ production through the activation of the Rho-family guanosine triphosphatase (GTPase) Ras-related C3 botulinum toxin substrate 1 (Rac1), which stimulates NADPH oxidase activity [42]. However, it is still unclear how blue light increases ROS production in macrophages.

## 5. Conclusions

The results of this study showed that daily exposure to blue light for 1 year induced skin cancer. Neutrophil NETs and type 1 macrophages play an important role in the induction of skin cancer. However, when type 1 macrophages persist, cancer declines because immunity is enhanced. Therefore, cancer cells shift macrophages to type 2, which has a cancer-proliferating effect. This study did not examine the shift from type 2 to type 1 macrophages, and further detailed studies are needed. It is also not well understood whether NETs actually play a role in cancer proliferation in neutrophils. Therefore, the effect of blue light on neutrophil NETs also needs further investigation. There are many ways to prevent blue light. In modern times, we use computers and smartphones on a daily basis for a long time. Therefore, use blue light cutting glasses when using screens at night. Also, since the blue light of the sun affects the body clock, it is recommended to be exposed in the morning, but it may be better to use blue light cutting glasses and blue light cutting base make-up outside in the morning. However, many of the effects of blue light on living organisms are unknown, and further research is required, including on methods of protection.

## Figures and Tables

**Figure 1 biomedicines-11-02321-f001:**
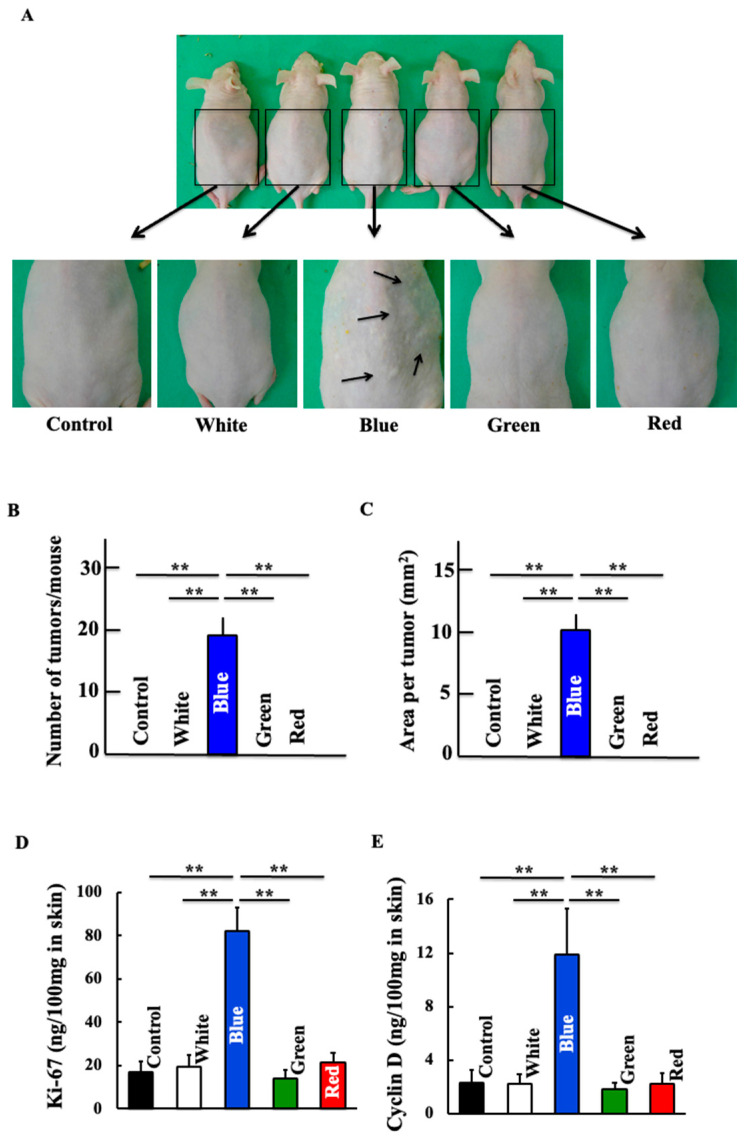
Effects of blue light irradiation on nonmelanoma skin cancer. (**A**) Representative example of each group showing macroscopic photographs of the dorsal skin. The number of tumors (**B**) and area per tumor (**C**) are shown. Dorsal skin levels of Ki-67 (**D**) and Cyclin D (**E**) were measured. Values are expressed as the means ± standard deviation (SD) derived from five animals. ** *p* < 0.01.

**Figure 2 biomedicines-11-02321-f002:**
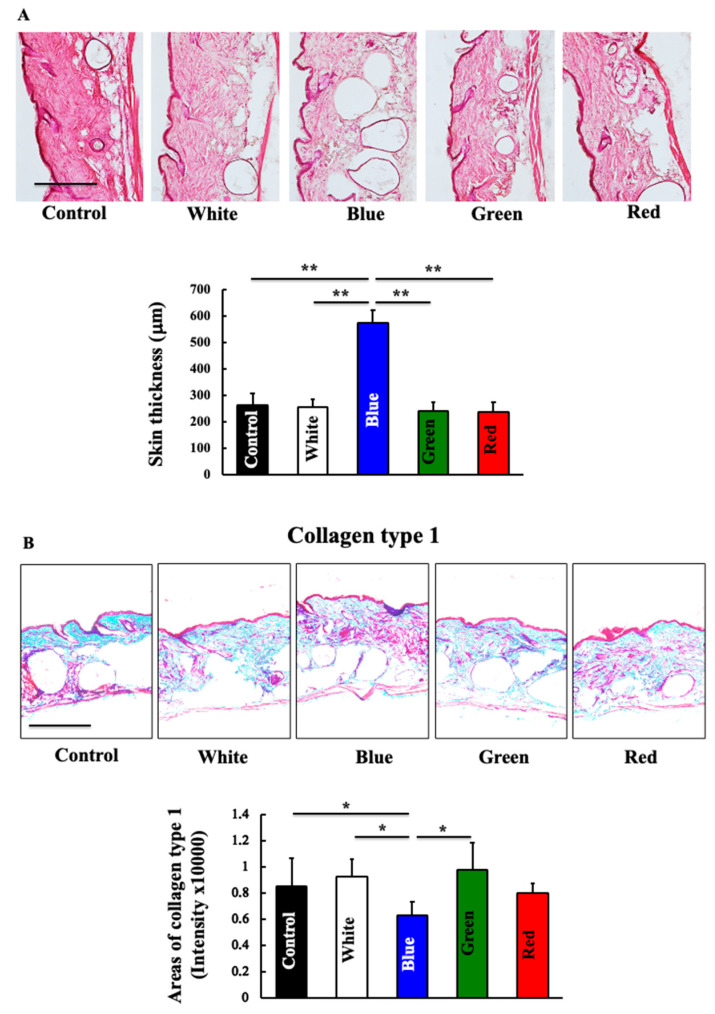
Effects of blue light irradiation on the skin thickness and on the expression of collagen. At the end of this study, we measured the skin thickness (**A**) and expression of collagen (**B**) in the dorsal skin of hairless mice. The values are expressed as means ± SD derived from five animals. * *p* < 0.05; ** *p* < 0.01. Scale bar = 100 μm.

**Figure 3 biomedicines-11-02321-f003:**
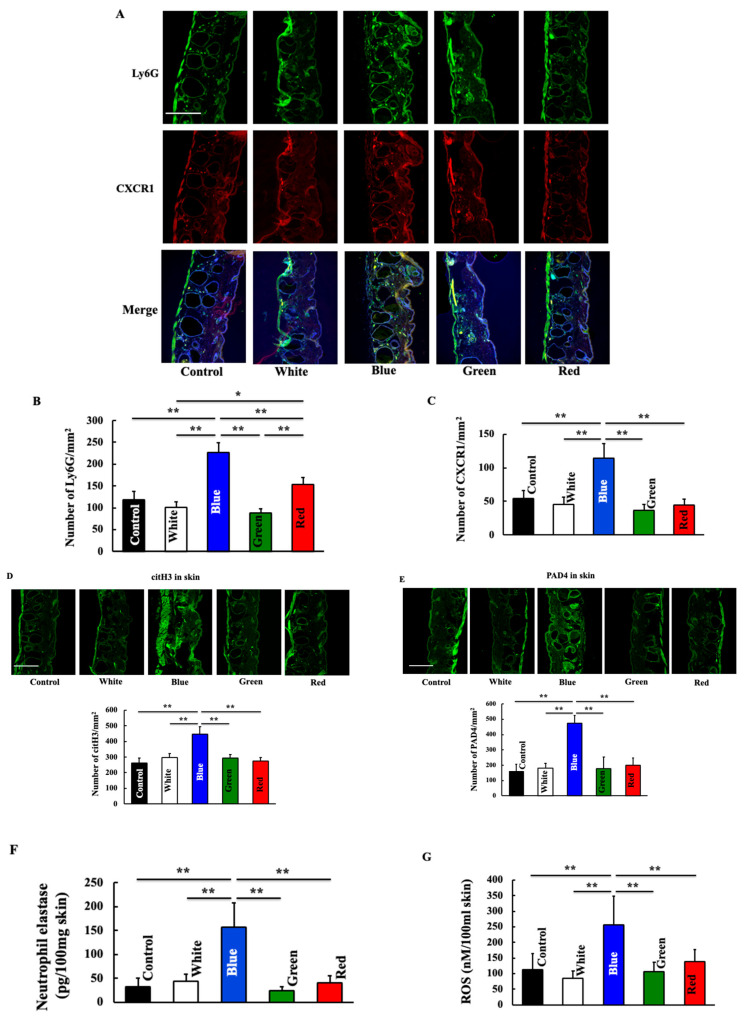
Effect of blue light irradiation on the expression of Ly6G (**A**,**B**), CXCR1 (**A**,**C**), citH3 (**D**), and PAD4 (**E**) and on skin levels of neutrophil elastase (**F**) and ROS (**G**). We performed immunostaining to examine the expression of Ly6G, CXCR1, citH3, and PAD4. Skin levels of neutrophil elastase and ROS were measured using a commercial kit. The values are expressed as means ± SD derived from five animals. * *p* < 0.05; ** *p* < 0.01. Scale bar = 100 μm.

**Figure 4 biomedicines-11-02321-f004:**
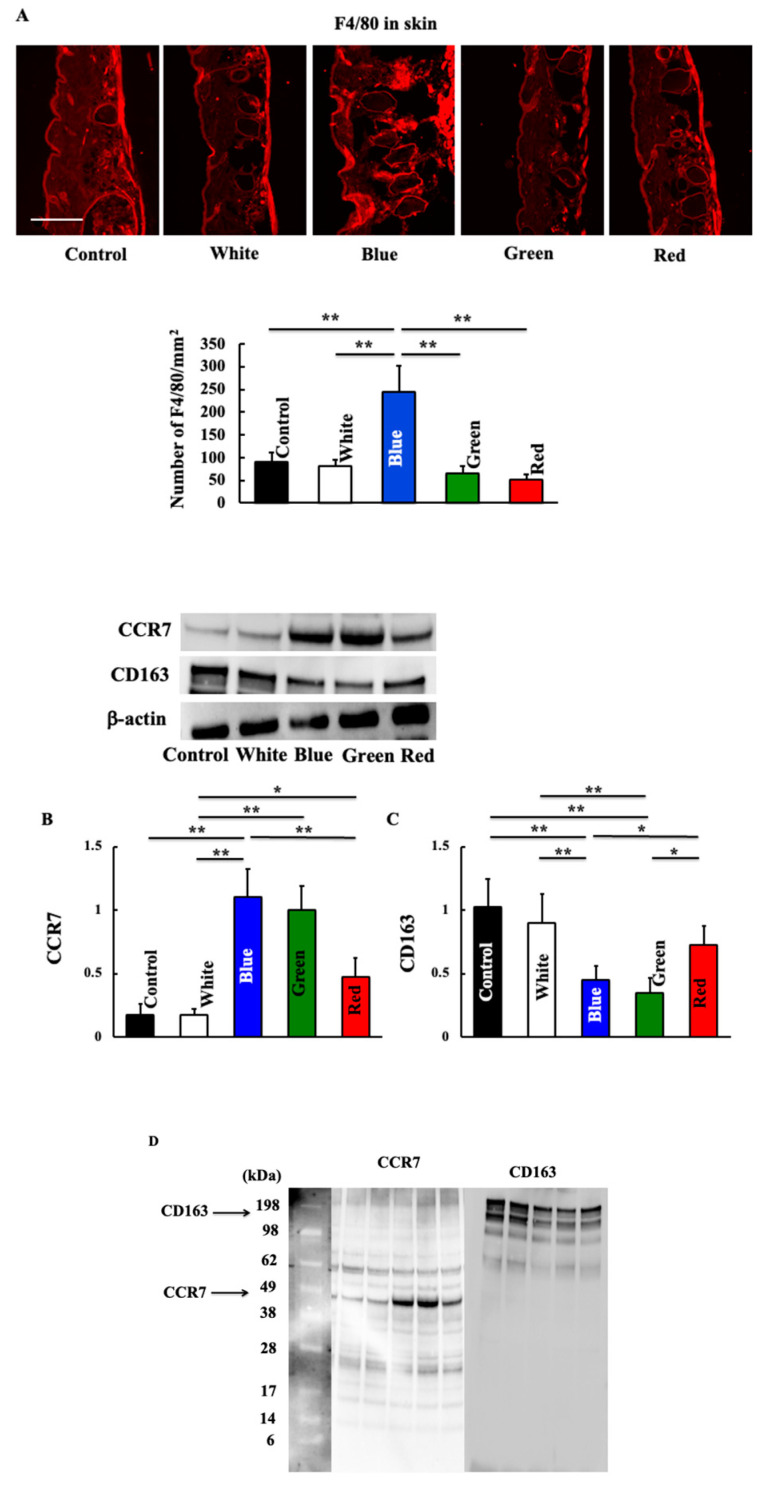
Effect of blue light irradiation on the expression of F4/80 (**A**), CCR7 (**B**), and CD163 (**C**) and on plasma levels of IL-10 (**E**), TGF-β (**F**), IL-6 (**G**), and IL-23 (**H**) after the final irradiation. We performed Western blotting to examine the expression of CCR7 and CD163 in the skin, and immunostaining to examine the expression of F4/80. Western blot diagram of CCR7 and CD163 with molecular weight markers (**D**). Plasma levels of IL-10, TGF-β, IL-6, and IL-23 were measured using an ELISA kit. The values are expressed as means ± SD derived from five animals. * *p* < 0.05; ** *p* < 0.01. Scale bar = 100 μm.

## Data Availability

Data are available within the article.

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
