# Peer review of "Induction of Skin Cancer by Long-Term Blue Light Irradiation"

_biomedicines, 2023, doi:10.3390/biomedicines11082321_

Round 1

Reviewer 1 Report

This is an interesting idea for a study that aims to investigate the effects of chronic blue light exposure on the skin of hairless mice. The results are somewhat intriguing but there is some essential information missing and a few inconsistencies/weaknesses in the data. It is particularly important that the evidence is put beyond doubt because the findings are of potentially high public impact, affecting people's understanding of the risks associated with blue light exposure. Specific comments:

(1) The introduction makes a reasonable attempt of setting the study into the wider photobiological context. However, there is a wealth of evidence that blue light can cause direct and indirect effects on skin including via induction of ROS production through photosensitiser-mediated reactions and this context should have been included in both the introduction and the discussion.

(2) Some important information about the basic experimental set up is lacking. For example, the identity of the specific mouse strain used in this study was not provided. Also, what kind of light source were the control mice exposed to? How intense was it? The white LEDs used as a control have a substantial blue spectral peak at around 450 nm. The authors should address the apparent lack of a biological effect of this blue light on the mice. Additional to this point, why was the white light used less intense than the other sources?

(3) In the data shown in Figure 1A, the expanded images don't seem to be from the matching images they are linked to above. The tumours are also quite difficult to see in the image. How were they confirmed to be NMSC, rather than a non-malignant lesion e.g. dermal cysts? There is an overall lack of descriptive analysis of this experiment e.g. at which stage of the 12 month treatment did tumours first start to appear?

(4) In Figure 1B proliferative markers are measured from samples of skin - in the blue light treated mice was this small sample (10 mg) take from the normal adjacent skin or was it from tumours? This point applies to subsequent experiments as well.

(5) The data in Figure 2 are claimed to show increased skin thickness and decreased collagen in the blue light treated mice. Firstly, there are some issues with how the data have been presented e.g. the  image of the histological section of skin from the blue light treated mouse in Figure 2A has been cropped too tightly and the full extent of the dermis isn't shown. There also seem to be increased amounts of dermal cysts in this section compared to the others - is this representative? The images in Figure 2B, which seem to show equal amounts of cysts between the various skin samples, suggest not. The graph shown in Figure 2A suggests that the skin in blue light treated mice is more than twice as thick as the skin in the other mice. However, this is not supported by the histology images shown in Figure 2B, where the thickness of the skin in the blue light treated mouse seems to be largely the same as in the others. There is also a lack of detail on how areas of collagen staining were measured in ImageJ - it would be helpful to have this included. Finally, the title for this figure "Effect of blue light irradiation on nonmelanoma skin cancer of dorsal mouse skin" does not reflect the data shown.

(6) The data shown in Figure 3 also lack clarity in how they've been presented. Again, are these samples taken from the "tumour" sites or the normal skin?What does "number of Ly6G" mean in the context of these data? Number of puncta? How were these counted/evaluated - was a cutoff for area/intensity used? Overall, there's a lack of clarity in the methodology for this and like all of the immunofluorescence data in this manuscript there is no DAPI staining to distinguish acellular material from cellular. The graph for Ly6G suggests that control skin has similar amounts to all the other skin samples apart from the blue light treated, but from the image it looks like there are more than in all of the other samples (apart from blue light treated). Other data in this figure purport to show NETosis but the IF images aren't 100% convincing - citH3 and PAD4 staining doesn't seem to be very specific (citH3 looks like it's in the epidermis in the blue light treated skin) and doesn't seem morphologically consistent with formation of NETs. Co-staining with DAPI/PI and MPO would have given further confidence that neutrophils were being stained.

(7) The evidence presented in Figure 4 for a shift from an M1 to M2 macrophage profile is quite weak. CCR7 is not a specific marker for pro-inflammatory macrophages, it's expressed in a variety of immune cells. The immunostaining of F4/80 seems to be quite non-specific and without a second stain, it's not clear that it's definitely due to macrophages - the image doesn't seem to show individual puncta that would be indicative of macrophages and so it's particularly unclear how "number of f4/80 per mm2" was measured in this case (again the absence of DAPI staining doesn't help).

Overall, this is an interesting premise for a study but the data are quite weak and do not support the conclusions that have been made. There is also a question of whether being kept for a year under blue, green or red light would cause stress in the test animals (which might explain increase in inflammatory markers) and how biologically relevant this model is to human health. Under what possible circumstances would a human being find themselves exposed to narrowband blue light for 12 hours a day, every day for a year?

Author Response

Response to the Reviewers’ comments

To Reviewer 1

We would like to thank the reviewer for appreciation of our work.

Thank you very much for your valuable comments on this paper. As the reviewer says, this paper does not clarify much about the details. In addition, immunostaining seems to be insufficient. It is difficult to add further analysis without a sample at this time. We are currently conducting long-term irradiation, and we plan to analyze the mechanism in more detail, keeping in mind what you pointed out.

 It has been reported that skin aging is induced by blue light. Since skin cancer is also induced by skin aging or aging, it is possible that long-term exposure to blue light may cause skin cancer.

  1. Thank you for your comment. We have redrawn Figure 6.

(Introduction: p.2, lines 11 - 15):

Also in the skin, photoaging consists of a decrease in skin elasticity in small pimples, lentigines, and mottled pigmentation. Blue light has been shown to induce photoaging in vitro and in vivo, specifically causing oxidative stress, damaging the skin barrier and promoting persistent skin pigmentation [16, 17, 18].

  1. Avola R., Graziano A.C.E., Pannuzzo G., Bonina F., Cardile V. Hydroxytyrosol from olive fruits prevents blue-light-induced damage in human keratinocytes and fibroblasts.J. Cell. Physiol.2019;234:9065–9076. 
  2. Mann T., Eggers K., Rippke F., Tesch M., Buerger A., Darvin M.E., Schanzer S., Meinke M.C., Lademann J., Kolbe L. High-energy visible light at ambient doses and intensities induces oxidative stress of skin-Protective effects of the antioxidant and Nrf2 inducer Licochalcone A in vitro and in vivo.Photodermatol. Photoimmunol. Photomed.2020;36:135–144.
  3. Austin E., Huang A., Adar T., Wang E., Jagdeo J. Electronic device generated light increases reactive oxygen species in human fibroblasts.Lasers Surg. Med. 2018;50:689–695.

(Discussion: p.9, lines 26 - 31):

In addition, blue light inhibits fibroblast proliferation and induces suppression of type I collagen expression and upregulation of MMP-1 [4,5,6]. And JUN, TGF and EGFR signaling pathways have been reported to influence blue light-induced skin aging [38]. In this study, immune cells such as neutrophils and macrophages were also involved in this transduction pathway, and it was thought that a more complex network was formed.

  1. Ge G, Wang Y, Xu Y, Pu W, Tan Y, Liu P, Ding H, Lu Y.M., Wang J, Liu W, Ma Y. Induced skin aging by blue-light irradiation in human skin fibroblasts via TGF-b, JNK and EGFR pathways. J Dermatol Sci. 2023; 28: S0923-1811(23)00150-0.

  1. Thank you for your comment.

(Materials and Methods: 2.1):

Mouse strain = Hos:HR-1

Light source of control mice = room lighting in the breeding room

White light contains blue light, but no histological or immunological effects were observed in this experiment.

  1. Thank you for your comment.

The reason for the diagnosis of NMSC in this study is that Ki-67 and Cyclin D increased. And tumors appeared around 10 months after irradiation.

  1. Thank you for your comment.

  Skin was collected from the tumor site.

  1. Thank you for your comment.

All photos are representative examples, and the most obvious ones have been selected and posted.

  (Materials and Methods: 2.2):

We have added an Image J measurement method.

Briefly, original files were converted to monochrome 8-bit file. Next, the threshold of luminous intensity was voluntarily established. Areas above the threshold were measured in each sample. These areas were defined as “intensity” in this study.

  We fixed the figure legend.

  (Figure 2):

Effects of blue light irradiation on the skin thickness (A) and on the expression of Collagen type I (B).

  1. Thank you for your comment.

    The number of Ly6G was counted by sorting the ones with constant fluorescence intensity by Image J.

  As reviewer said, co-staining with DAPI/PI makes it even clearer. This research does not use DAPI, which will be a subject for the next time.

  1. Thank you for your comment.

As the reviewers say, evidence of M1 to M2 shift in macrophages is weak, making the results of this trial fragile. We plan to update this study and examine macrophages in more detail.

Reviewer 2 Report

This interesting study demonstrates that long-term exposure of nude mice to blue but not white, green, or red light induced skin cancer, and the presence of neutrophils and inflammatory macrophages in the skin, and enhanced the levels of citH3 and PAD4 in the neutrophils, suggesting the possibility of NETosis. These findings suggest that continuous exposure to blue light might be dangerous as it can be a cancerogenic agent.

Abstract: is the term “expression” optimal with respect to the abundance of neutrophils and macrophages? It is usually used to denote gene activity (the level of transcript)

Remarks:

2.1. The authors provided the same exposure conditions for various light types (40 kJ/m2). How was it measured or was this value declared by the manufacturer?

2.1. “The entire bodies of mice were exposed to each LED light daily”. Was the experimental light the only source of illumination? Was the light source placed above the animals to secure uniform exposure?

2.1. “…were exposed daily”; does it mean 12-h exposure?

2.2.: were the mouse samples reactive with mouse monoclonal anti-lymphocyte antigen 6 complex locus G6D?

2.3. “Western blotting of dorsal skin”, whole skin can not be blotted, perhaps “… of dorsal skin proteins”

2.4. “The samples were then homogenized at 1,500 x g and 4 °C for 15 min”, do the authors mean conditions of homogenization or of subsequent centrifugation?

Author Response

Response to the Reviewers’ comments

To Reviewer 2

We would like to thank the reviewer for appreciation of our work.

Abstract: Since neutrophils and macrophages migrate, "migration" seems appropriate here.

2.1. Thank you for your comment. We measured the amount of energy using a    light analyzer LA-105 (NIPPON MEDICAL & CHEMICAL INSTRUMENTS Co., Ltd.).

2.1. Thank you for your comment. Fluorescent lights are used for regular rearing. The control group was irradiated with this fluorescent lamp. Also, the light is applied from above the animal.

2.1. Thank you for your comment. The whole body of mice was exposed to each LED. light daily (10 minutes a day) for one year.

2.2. Thank you for your comment. We rewrote.

2.3. Thank you for your comment. I corrected.

2.4. Thank you for your comment. This shows the centrifugation conditions.

Reviewer 3 Report

In this manuscript, Hiramoto et al. explored the potential risk of blue light irradiation for inducing skin cancer. This work is of great interest to readers in the skin cancer area. Some minor concerns are shown below before consideration of publication in this journal.

1. What kind of ROS is induced by blue light in dorsal skin? How was it generated? It's important for understanding the underlying mechanism.

2. How to prevent blue light induced skin cancer?

Author Response

Response to the Reviewers’ comments

To Reviewer 3

We would like to thank the reviewer for appreciation of our work.

  1. Thank you for your comment. I added blue light and ROS.

(Discussion: p.9, lines 32 - 45):

Furthermore, in this study, the amount of ROS produced was increased by blue light irradiation. ROS play an important role in tumor growth. Neutrophils produce ROS by activation of nicotinamide adenine dinucleotide phosphate (NADPH) oxidase. NADPH oxidase consists of two subunits, p22Phox and gp91phox, and contains all the electron transport chains necessary for the production of superoxide anion (O2- ) from NADPH, and converts O2- to hydrogen peroxide (H2O2) and singlet molecular oxygen (1O2) [39,40]. This suggests that blue light may activate NADPH oxidase. Macrophages are the first immune cells that exert direct antibacterial effects by producing ROS [41]. In macrophages, brain-specific angiogenesis inhibitor 1 (BAI1), an adhesion family member of G protein-coupled receptor (GPCR)s, promotes O2- production through activation of the Rho-family guanosine triphosphatase (GTPase) Ras-related C3 botulinum toxin substrate 1 (Rac1), which stimulates NADPH oxidase activity [42]. However, it is still unclear how blue light increases ROS production in macrophages.

  1. Vignais, P.V. The superoxide-generating NADPH oxidase. CMLS. 2002, 59, 1428-1459.
  2. Miyata, K.; Tamura, M.; Sumimoto, H. Molecular mechanism for production of reactive oxygen species by neutrophils. Inflamm. Regen. 2023, 25, 113-117.
  3. Epelman, S.; Lavine, K.J.; Randolph, G.J. Origin and functions of tissue macrophages. Immunity. 2014, 41, 21-35.
  4. Billings, E.A.; Lee, C.S.; Owen, K.A.; D’Souza, R.S.; Ravichandran, K.S.; Casanova, J.E. The adhesion GRCR BAI1 mediates macrophage ROS production and microbicidal activity against Gram-negative bacteria. Sci. Signal.2016, 9, ra14.
  5. Thank you for your comment. I added the blue light protection method in the conclusion section.

(Conclusion: p.10, lines 5 - 11):

There are many ways to prevent blue light. In modern times, we use computers and smartphones on a daily basis for a long time. Therefore, use blue light cut glass when using at night. Also, since the blue light of the sun affects the body clock, it is recommended to be exposed in the morning, but it may be better to use blue light cut glasses and blue light cut base make-up outside the morning. However, many of the effects of blue light on living organisms are unknown, and further research is required, including methods of protection.